# Subjective well-being during the 2020–21 global coronavirus pandemic: Evidence from high frequency time series data

**Roberto Stefan Foa[1], Mark Fabian [2]\*, Sam Gilbert[1]**

**1** Department of Political Science and International Studies, Bennett Institute for Public Policy, University of Cambridge, Cambridge, United Kingdom, **2** Institute for Social Change, University of Tasmania, Hobart, Australia

\* mark.fabian@utas.edu.au

## Abstract

We investigate how subjective well-being varied over the course of the global COVID-19 pandemic, with a special attention to periods of lockdown. We use weekly data from You-Gov's Great Britain Mood Tracker Poll, and daily reports from Google Trends, that cover the entire period from six months before until eighteen months after the global spread of COVID-19. Descriptive trends and time-series models suggest that negative mood associated with the imposition of lockdowns returned to baseline within 1–3 weeks of lockdown implementation, whereas pandemic intensity, measured by the rate of fatalities from COVID-19 infection, was persistently associated with depressed affect. The results support the hypothesis that country-specific pandemic severity was the major contributor to increases in negative affect observed during the COVID-19 pandemic, and that lockdowns likely ameliorated rather than exacerbated this effect.

## Introduction

The dramatic and widespread impacts of the COVID-19 pandemic make it imperative that we understand the efficacy of policy responses to it. Among the most prominent such policies are 'lockdowns'–mandated or voluntary stay-at-home and shelter-in-place-orders that promote social distancing and reduce the spread of the virus. The evidence to date supports the view that lockdowns were good for physical health in that they reduced excess mortality associated with the virus [1, 2]. However, the psychological effects of lockdowns remain unclear.

One of the most prominent such effects is upon subjective well-being (SWB), typically defined as a combination of experienced mood and evaluated life satisfaction, as well as feelings of meaning and purpose [3, 4]. Lockdowns could be expected to impact SWB negatively by, for example, reducing social interaction [5], increasing the burden of child care [6], or by exacerbating stress and boredom [7]. Conversely, lockdowns could also improve subjective well-being by, among other things, eliminating the necessity for long commutes to work [8], allowing more time to socialise with close family members [9], and alleviating anxieties caused by the spread of COVID-19.

**Data Availability Statement:** The raw data cannot be shared publicly as they are the commercial property of YouGov. However, the time series of the data is published weekly here and is sufficient

to verify our analysis: https://yougov.co.uk/topics/science/trackers/britains-mood-measured-weekly We will make all do files available to any researcher who wishes to replicate our analysis.

**Funding:** The author(s) received no specific funding for this work.

**Competing interests:** The authors have declared that no competing interests exist.

The majority of empirical studies that examine SWB under lockdown compare the results of surveys conducted before the global coronavirus pandemic with results from surveys that were fielded after lockdowns had been introduced. They typically find a deterioration in SWB and/or mental health between the two surveys. Yet this leaves ambiguous the cause of the observed decline. It could be that the imposition of lockdown measures had a negative effect on SWB. However, it could also be that the pandemic itself depressed wellbeing, for example as a result of fear or ill-health among vulnerable populations, and that lockdown policies had an attenuating role.

This is frustrating because, from a policy perspective, we want to understand whether lockdowns in response to pandemic outbreaks worsen or improve citizens' overall sense of wellbeing. Numerous commentators, including prominent politicians in the United States (US) and the United Kingdom (UK), have argued that lockdowns should be curtailed or ended owing to their negative impacts on SWB and mental health. Yet if the negative effects observed are principally driven by fear of the pandemic, bereavement, or the lasting symptoms caused by COVID-19 infection, then ending lockdowns prematurely may have the opposite of the intended outcome.

In this study, we shed light on this question using high-frequency observational data covering the entire duration of the pandemic, including multiple separate coronavirus waves, as well as government response measures to them. Specifically, we use two years of weekly survey data from YouGov's Great Britain mood tracker, together with two years of daily global search data from Google Trends from six countries, to provide insight into whether lockdowns in response to pandemic outbreaks can be expected to worsen or improve SWB. To date, ours is the first study that provides a comprehensive overview of the 2020–21 period, across multiple waves of infection and policy response. Our data includes both cases where lockdown measures were introduced yet prevented a widespread epidemic via community transmission (such as New Zealand and Australia in the spring of 2020), and instances where lockdowns were eased according to original schedules despite the onset of a new coronavirus wave (such as the United States in the summer of 2020, and the United Kingdom in summer 2021). While identification of the independent causal effect of pandemics and lockdowns upon SWB is necessarily frustrated by endogeneity, the existence of cases where pandemic waves and lockdowns occurred to some extent separately of one another offers an improved basis for prediction over single-country studies using individual pre- and post- surveys as evidence.

Observation of the descriptive trends in affect reveals pronounced structural breaks in a negative direction following pandemic outbreak, and a positive direction following the imposition of lockdowns. These observations are mirrored by the results of statistical modelling, which show a consistent negative association between affect and pandemic severity, and a strong and steady recovery in affect after the first few days of lockdown. This association between lockdowns and affect is robust to controls for hedonic adaptation [10] and progress in containing the virus outbreak. Following the imposition of lockdown we also observe an especially strong recovery trend among the most vulnerable population (the over 65s), which suggests that reduced ill-health and anxiety among such groups may be a plausible explanation. While SWB is not identical to mental health, the two concepts are closely related statistically [11], suggesting that we should predict a worsening of mental health during pandemic outbreaks, and improvements following the implementation of lockdowns in response. Together with other recent studies, showing for example a decline in suicide rates across a large sample of countries during the pandemic [12], our results provide a valuable additional input into contemporary and future policy debates over when to ease lockdown restrictions in pandemics for the sake of SWB.

## Literature review

Studies published around the time lockdowns were initially implemented in western countries raised concerns about the possibility of negative mental health effects related to SWB, including loneliness, depression, and suicide [13, 14]. There are several reasons why lockdowns could have such negative impacts. First, being quarantined reduces social interaction, an important correlate of SWB [5]. Secondly, the dramatic nature of lockdown policies could even further exacerbate stresses and anxieties concerning the threat posed by the pandemic [15]. In addition, similar feelings could be fuelled by the challenges associated with balancing work and home life in lockdown conditions, especially in small and/or crowded households [6, 16]. With many schools closed, parents were burdened with the duty of home schooling, while several studies note an uptick in reports of domestic violence [17].

On the other hand, lockdowns are a measure taken in response to a pre-existing threat: namely, an ongoing or impending pandemic. As of the latest count, 0.8% of the entire elderly (75 and above) population of the United States has died while diagnosed with COVID-19 during 2020–21, with a case fatality ratio of 17% [18]. Against this background, lockdowns could have a positive effect on well-being insofar as they provide a forceful policy response to the pandemic that enhances people's sense of security [19]. If lockdowns bring down the prevalence of the virus faster than voluntary self-isolation, they might also reduce the time individuals need to self-isolate and the associated negative psychological effects. In addition, many lockdowns were introduced with substantial economic support programs. In the UK for example, rental payments were delayed, debts were temporarily forgiven, welfare payments were increased, and the stringency of welfare under the British Government's universal credit scheme was relaxed. Economic stress can lead to mental distress, and economic security is a well-established source of SWB [20]. As such, lockdowns and their associated economic support may boost SWB, especially among groups that are typically under economic and mental strain.

Some empirical studies of the effects of lockdown appear to support the view that lockdowns were bad for well-being [21]. Studies using the United Kingdom Household Longitudinal Study, which is based on a probability sample, found that the prevalence of clinically significant levels of mental distress, measured using the General Health Questionnaire (GHQ-12), had increased by around 8 percentage points one-month into the UK lockdown compared to previous years [22, 23]. Similar results were found in similar studies from New Zealand [19] and the United States [24, 25]. Early studies from China also found modest declines in SWB and worsening psychological distress [26–28]. However, other empirical studies have found mixed effects varying by aspect of SWB and mental health, and heterogeneous effects by demographic characteristics [29]. One analysis using data from the COVID Social Study in the UK, which commenced after lockdown was in place, found that anxiety and depression did not worsen during lockdown [30]. Another study using the same data set found that lockdowns exacerbated loneliness among the already lonely but reduced it among the least lonely [31]. Similar, though milder, effects on loneliness were observed in a separate study from the United States [32]. A French study found that SWB, operationalised using questions about whether respondents felt nervous, low, relaxed, sad, or happy, improved during lockdown, except among Parisians [33].

These studies are not, however, able to provide insights into the differential effects of the pandemic and subsequent lockdowns as separate albeit related events. They typically rely on measures of SWB and/or mental health taken well before the onset of the pandemic, and then follow-up surveys administered after lockdowns were introduced. Those that rely on data collected after lockdowns began have no baseline against which to measure changes in SWB and

mental health, and must instead rely on people's own assessments. Meanwhile, longitudinal studies that do possess a baseline typically do not have observations in the period between the advent of the pandemic and the introduction of lockdowns, and hence estimate only their joint impact. Yet a deterioration observed in the second period could be caused by pandemic onset and then be further exacerbated or potentially ameliorated by lockdowns. It is difficult if not impossible to separate the causal effects of the pandemic from those of lockdowns analytically, but precisely because the latter are introduced as a response to the former it is important that we do not lump these two events together as a single phenomenon. If SWB tends to decline with pandemic onset and improve when lockdowns are introduced in response, then avoiding lockdowns or ending them prematurely because we are worried about their psychological effects may have the opposite of the intended outcome.

To our knowledge, there are only two papers on the impacts of lockdowns on SWB with data capable of at least temporally differentiating these two factors. The first is Zacher and Rudolph (2020) [34], who use a nationally representative, longitudinal sample of around 1,000 employed Germans surveyed on four separate occasions–December 2019, when the first COVID-19 cases were reported in China; March 2020, around the time of the first COVID-19 death in Germany; and then again in April and May 2020 during the initial months of the first national lockdown. Respondents were asked about their life satisfaction on a scale from 1–7, and about a host of a affective states drawn from the short form Positive and Negative Affect Schedule [35]. The positive affects were: inspired, alert, excited, enthusiastic, and determined; while the negative affects were afraid, upset, nervous, scared, and distressed. Using growth curve analysis, the authors find systematic declines (i.e. worsening) in life satisfaction and positive affect associated with the imposition of lockdown, but also declines (i.e. improvements) in negative affect. A second paper capable of providing some insights into the differential effects of the pandemic and lockdowns is Daly and Robinson (2021) [36]. They use a representative sample of US adults taken fortnightly from March to June 2020. They find an increase in psychological distress in March through April, but then a recovery. While not conclusive nor causal, these results would incline us to predict a deterioration in SWB at pandemic onset, and then a moderation or improvement in SWB following the imposition of lockdown.

Our analysis has notable strengths and weaknesses relative to these papers. We contribute insights from the experience of lockdown in the UK using weekly data from YouGov, and daily data from six English-speaking countries using Google Trends. We also have a larger sample, higher-frequency data, and a longer time series, covering the entire period from July 2019 to June 2021. These allow us to estimate time-series and multi-level models to complement Zacher and Rudolph's growth-curve analysis. However, our sample is cross-sectional rather than longitudinal, which means we can only assess changes in SWB in the aggregate, rather than at the individual level. Our analysis also relies overwhelmingly on mood variables, rather than a more complete set of SWB questions.

## Data and methods

Our analysis utilises weekly data from YouGov's Great Britain Mood Tracker poll and daily reports from Google Trends. For the United Kingdom, our sample covers both the 6 months before the pandemic as well as its first 18 months, which includes two major lockdown periods (April to June 2020 and January to April 2021), one briefer 'mini-lockdown' (November 2020), and several coronavirus waves (the initial wave in the spring of 2020, the 'alpha' variant waves in October 2020 and January 2021, and the spread of the 'delta' variant in summer 2021). For the broader global sample of countries using daily data from Google Trends, our sample period includes instances where lockdowns occurred without uncontrolled community

transmission leading to a full epidemic (New Zealand and Australia in 2020), and also corona-virus waves that occurred in a context of reduced or continued easing of lockdown rules (in the United States in the summer of 2020, and across several countries in 2021).

### The Great Britain weekly mood tracker survey

YouGov is one of the world's most reputable polling and market research companies, and the source of the largest cross-country COVID-19 global tracking survey currently in use among public health researchers [37]. From June 2019 to date, they have also surveyed the feelings and well-being of more than 200,000 respondents across England, Scotland and Wales, in weekly samples of between 1,890 and 2,071 individuals. Respondents are drawn from a larger panel of over 1 million British participants recruited by YouGov since 2000, representing more than 1.5% of current UK population, from which individuals are selected for each cross-sectional poll so as to be representative by age, gender, social class and education. Sampling is continuously assessed for reliability and accuracy based on disaggregated census returns and predictive accuracy in national elections. As of December 2020, when the latest individual-level dataset was provided to us by YouGov, a total of 154,053 respondents had completed the Mood Tracker survey, with additional surveys continuing to be conducted on a weekly basis. Individuals were asked to complete a shortened variant of the Profile of Mood States (POMS) battery [38]. This asks whether participants had experienced any from a list of positive and negative mood states during the past week: happiness, sadness, apathy, energy, inspiration, stress, optimism, boredom, contentment, loneliness, and fear. In addition, a total of 13,954 respondents from within these surveys also completed a variant of the 11-point Cantril Scale, which reports life satisfaction on a 0 to 10 scale, with 0 being the worst possible level, and 10 the best possible. Unfortunately, these life satisfaction responses are all from April 2020, but we can use them to predict life satisfaction for the whole period over which we have mood data (see below).

### Google trends

To validate the YouGov Great Britain weekly mood tracker results and facilitate cross-country comparisons, data was collected from Google Trends, which enables the relative popularity of Google searches to be analysed. Google Trends allows for a comparison of both search queries and 'topics' (clusters of related queries), and has previously been applied to research questions in the elds of public health [39, 40], economics [41–43], and political science [44], among others. Data for Google Trends topics was acquired for six English-speaking countries during the period from 30 June 2019 to 21 June 2020, corresponding to matching affective states in the YouGov weekly mood tracker: stress ('psychological stress'), boredom, frustration, sadness, loneliness, feeling scared ('fear'), apathy, happiness, contentment, energy, inspiration ('artistic inspiration'), and optimism.

### Descriptive statistics and trends

We begin our analysis with an overview of how the prevalence of specific mood states changed in the UK during different stages of the COVID-19 pandemic. Positive affect states–happiness, energy, inspiration, optimism, and contentment–show a very similar pattern (Fig 1). While there is clear stochasticity in the pre-pandemic trends, levels were relatively stable before the crisis. They then fell sharply during the virus breakout in March before reverting higher following the stay-at-home order.

Negative mood states, in contrast, show more heterogeneous trends. In the period of the pandemic breakout from 5 March to 26 March 2020, feelings of fear, stress, sadness, and

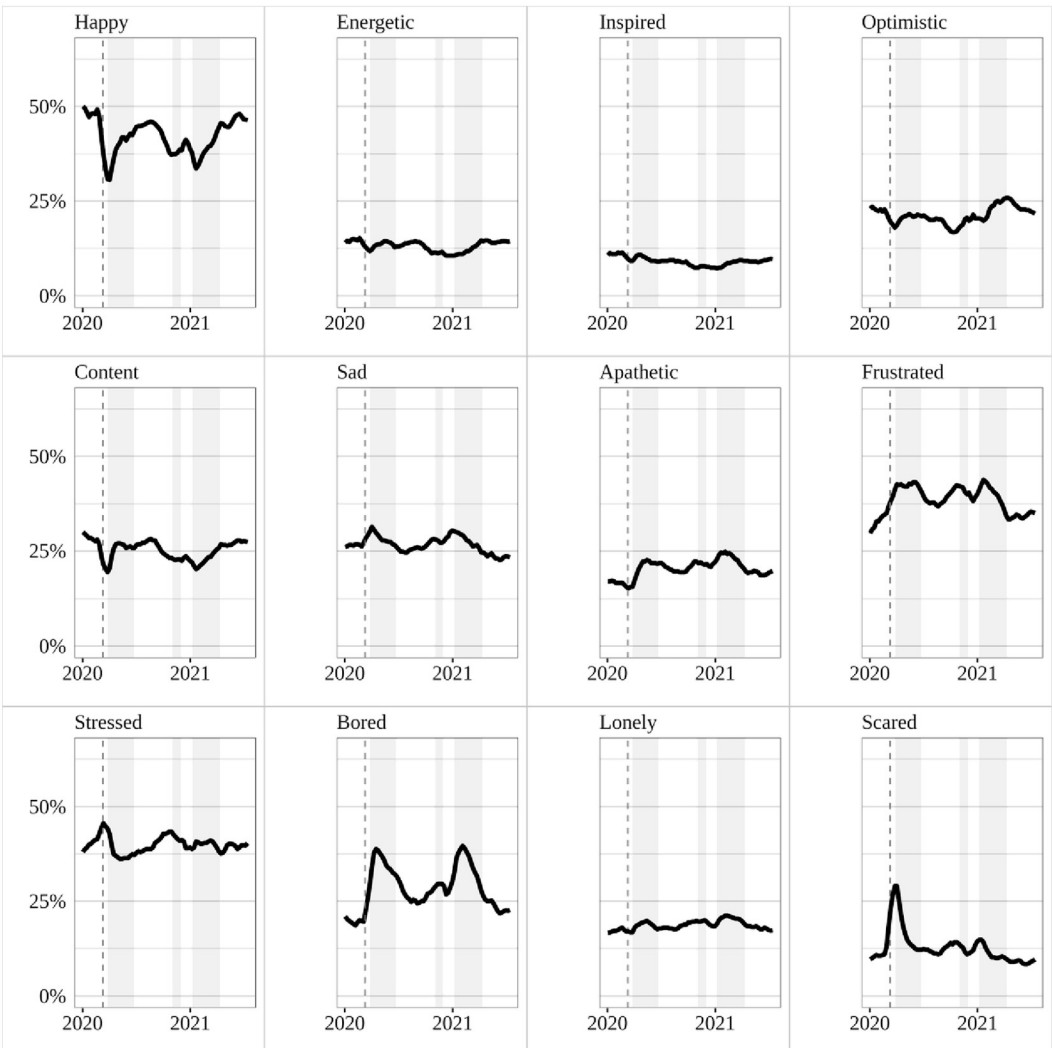

**Fig 1. YouGov mood tracker, 2019–2021: Descriptive trends.** Mean scores by week; rolling averages. The start of the pandemic is indicated by the dashed lines, while lockdown periods are indicated by the shaded portions on charts. Source: YouGov.

frustration all rose, presumably as individuals became attuned to the risks facing their health and livelihoods. However, there were also statistically significant falls in apathy (95% c.i. = -8.16% < x < -4.52%) and loneliness (95% c.i. = -5.92% < x < -2.05%). The first month of lockdown brought substantial falls (i.e. improvements) in fear and stress. Indeed, stress levels after one month of lockdown reached their lowest levels of the year. In addition, sadness also fell after reaching a peak during the first week of lockdown. These trends speak to the mortality risks associated with the virus and the lockdowns having a calming effect through their reduction in that risk. However, feelings of loneliness, apathy, frustration and boredom spiked higher, and while boredom, sadness, and loneliness fell back again in the second month, perhaps as people adapted to living at home, frustration continued upwards. This suggests that while we should expect lockdowns to coincide with an amelioration in moods that worsened during pandemic outbreak, some mood states will decline with lockdown. Taking all negative affect items together, negative affect rose sharply with the outbreak of the pandemic, and then

continued to rise, albeit much more slowly, after the imposition of lockdown. Immediately prior to the pandemic (February 2020), the average negative mood state response among respondents (across all negative mood states) was 23.8%. This rose to 27.6% in March and 30.1% in April, 2020. The pre/post pandemic increase is statistically significant at the $p < 0.001$ level.

On face value, these trends suggest an overall positive correlation between lockdown and mood, with positive affect recovering markedly during lockdown and increases in negative affect decelerating. However, due to the heterogeneity in trends across mood states, one can arrive at a more pessimistic conclusion depending on which mood states one regards as relatively important. To allow respondents to determine this weighting over mood states, we developed a summary index of 'affective life satisfaction' that estimates the independent association of each mood state with reported life satisfaction. We did this by regressing the individual mood states reported in the modified POMS question battery on the YouGov life satisfaction data from April 2020, and used this model to predict life satisfaction for all weeks for which mood data was available, thereby imputing life satisfaction in the manner of a wage equation [45]. This method weights each mood state by its contribution to reported life satisfaction, which is considered an effective global measure of SWB [3, 4]. All independent effects had the expected polarities, and coefficients for imputation models are shown Table A.1 in S1 Appendix. The largest effect magnitude for predicting life satisfaction was the mood state response for feeling 'happy', which accounted for 24% of the total variance in Cantril scale life satisfaction that could be explained by the mood state indicators. Feelings of loneliness accounted for a further 13% of explained variation, followed by sadness (13%), contentment (11%), stress (9%), optimism (8%), apathy (7%), fear (4%), frustration (4%), energy (4%), boredom (3%), and inspiration (2%).

The ALS index estimates that portion of life satisfaction that is due to respondents' positive and negative affective states. SWB consists of both 'experienced' well-being, which is made of affective states, and a cognitive component, typically referred to as 'evaluative' well-being [3, 4]. The ALS only directly captures the former. Nonetheless, it provides a reasonably close empirical approximation of overall life satisfaction: individual mood states could be used reliably to predict Cantril Scale life satisfaction at the individual respondent level (13,954 observations; R = 0.57), by sociodemographic group (48 observations, R = 0.88; see Fig A.2 in S1 Appendix), and almost perfectly in ALS-response clustered comparisons (63 observations, R = 0.99; see Fig A.1 in S1 Appendix). The cluster comparison involved organising the 13,954 individuals who answered both the profile of mood states battery and the Cantril scale (0–10) life satisfaction question into 63 groups using their scores on the affective life satisfaction measure rounded to one decimal place. Values ranged from 2.4 (the lowest group cluster) to 8.6 (the highest cluster). The mean average surveyed Cantril scale life satisfaction response for each group correlated almost perfectly (R = 0.99) with their mean affective life satisfaction scores ($R^2$ of 0.97).

Fig 2 shows the change in average affective life satisfaction in the UK from June 2019 to July 2021. This gives us a sense for the overall affect of British residents over the pandemic and the two major lockdown periods that occurred from March to July of 2020 and from January to April of 2021. A large and statistically significant drop in affect occurred before the implementation of lockdown measures during the period from Thursday 5 March, when the first diagnosed COVID-19 death in the United Kingdom occurred, to Thursday 26 March, when lockdown measures began. The low point for affect was recorded only three days after the announcement of the 'stay-at-home' order, and on the exact day that police enforcement measures came into effect, and thereafter affective life satisfaction rises steadily. A similar gradual decline in affect occurred concurrent with the spread of the Alpha variant from September

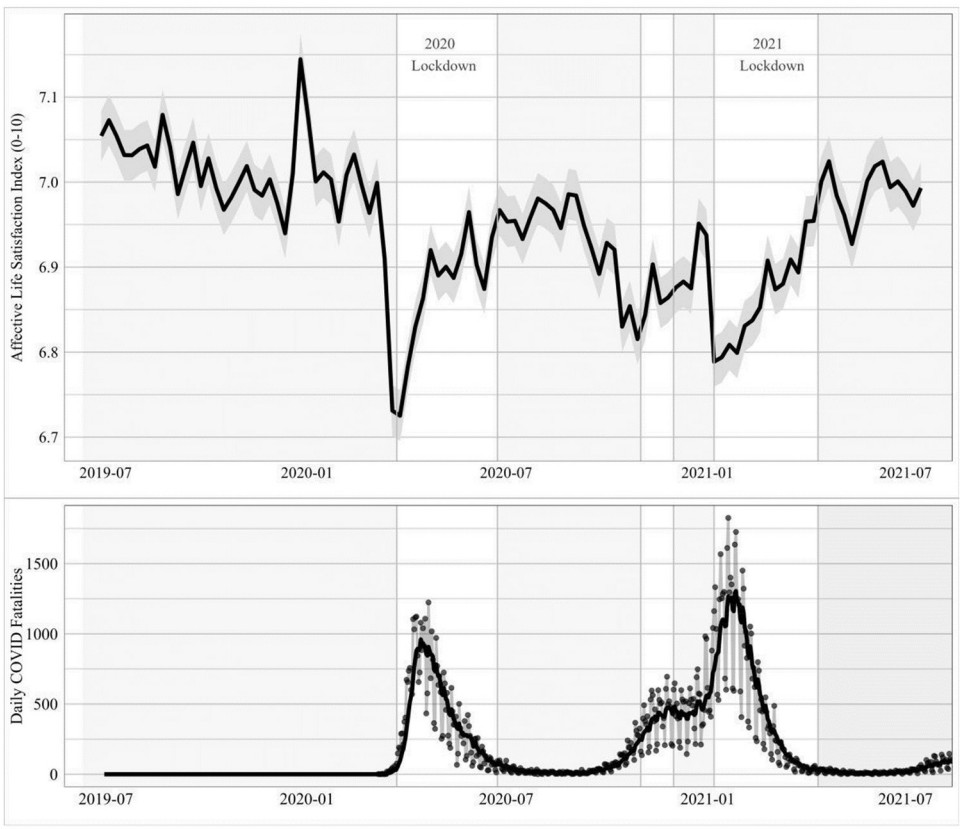

**Fig 2. Raw series trend: Affective life satisfaction, June 2019 to July 2021.** Mean scores by week, with 90% confidence intervals. Source: Affective life satisfaction index calculated from YouGov Great Britain mood tracker survey; COVID-19 data from Johns Hopkins University (2021).

2020 to January 2021, with affect reaching another low on the eve of the announcement of the second major lockdown at the start of 2021, where after it recovered steadily, returning to the pre-pandemic baseline by April of this year.

### Cross-country comparisons with search data

The geographic scope of the YouGov weekly mood tracker is limited to a single country. To enable cross-country comparisons, in June 2020 we supplemented British survey data from YouGov with Google Trends data on search-based equivalents of the affect measures for a wider range of cases during the initial months of the pandemic [46]. In this article we are able to extend this methodology to cover the entire two-year period from July 2019 to June 2021, as well as to confirm how well the original methodology has predicted subsequent YouGov survey observations. To determine how effectively changes in affect as measured in the YouGov data were proxied by Google Trends topics, Pearson's R correlations were calculated for each mood state measured and corresponding Google Trends topic during the 50-week period under observation. These results are shown in Table 1. With the exception of 'loneliness', Google Trends topics were found to be a reasonable proxy for negative moods, but a poor proxy for positive moods.

In order to confirm the validity of the data, 'Related Queries' were also qualitatively reviewed to check for the extent of false positives, i.e. search queries which are lexically related,

**Table 1. Mapping of YouGov mood states to Google trends topics.**

| YouGov Mood State | Corresponding Google Trends Topic | R value | Accepted as proxy? |
|---|---|---:|---|
| *Negative Affect* | | | |
| Stressed | Psychological Stress | 0.46 | Yes |
| Bored | Boredom | 0.85 | Yes |
| Frustrated | Frustration | 0.65 | Yes |
| Sad | Sadness | 0.55 | Yes |
| Lonely | Loneliness | 0.01 | No |
| Scared | Fear | 0.49 | Yes |
| Apathetic | Apathy | 0.44 | Yes |
| *Positive Affect* | | | |
| Happy | Happiness | -0.05 | No |
| Content | Contentment | -0.41 | No |
| Energetic | Energy | 0.19 | No |
| Inspired | Artistic Inspiration | -0.08 | No |
| Optimistic | Optimism | -0.32 | No |

Notes: R-values calculated for the 50 shared weekly affective state observations in both the YouGov Mood Tracker survey and weekly Google search data.

but do not imply the corresponding mood state. Google Trends describes the concept of Related Queries as follows: 'Users searching for your term also searched for these queries'. False positives partially explained the weakness of Google Trends topics as a proxy for positive mood states. For example, the topic 'energy' contained queries relating to gas and electricity suppliers, while the topic 'happiness' included queries relating to 'happy birthday', possibly reflecting a UK government public health campaign encouraging citizens to wash their hands for as long as it takes to sing 'Happy Birthday' twice. Of the negative moods, only the topic 'apathy' contained obvious false positives, though not sufficient to eliminate covariance between weekly apathy-related searches and surveyed apathy levels in the YouGov data. Related Queries for apathy included esoteric searches such as 'indifferent crossword clue', but also substantive queries largely related to mental self-help and diagnosis.

As Google Trends topics were a poor proxy for positive mood states, we developed our cross-country index using negative mood states only. To facilitate this analysis, we aggregate individual negative mood states in the YouGov data into a 'negative affect index' and the mood states in the Google Data into a 'negative affect *search* index'. The negative affect index takes average mentions from the list of possible negative states–sadness, apathy, frustration, stress, boredom, loneliness, and fear–making it analogous to the negative affect component of the widely used Positive and Negative Affect Scale (PANAS) [47]. The original index was calculated for a 50-week period from July 2019 to June 2020, and has been subsequently extended until June 2021, allowing us to compare 'in-sample' observations (those available at the time of index construction) with out-of-sample performance (observations made in the year since the index was designed). To construct the 'negative affect search index', we weight the Google Trends topics by their $R^2$ correlation coefficient with their matching YouGov survey mood state. At the time of the index construction in June 2020, the search-based negative affect index correlated highly ($R = 0.92$, $R^2 = 0.84$) with the sum of negative mood states reported in the weekly polling data series, a correlation that has held at a similar level during the 'out-of-sample' period in the following year (Fig 3).

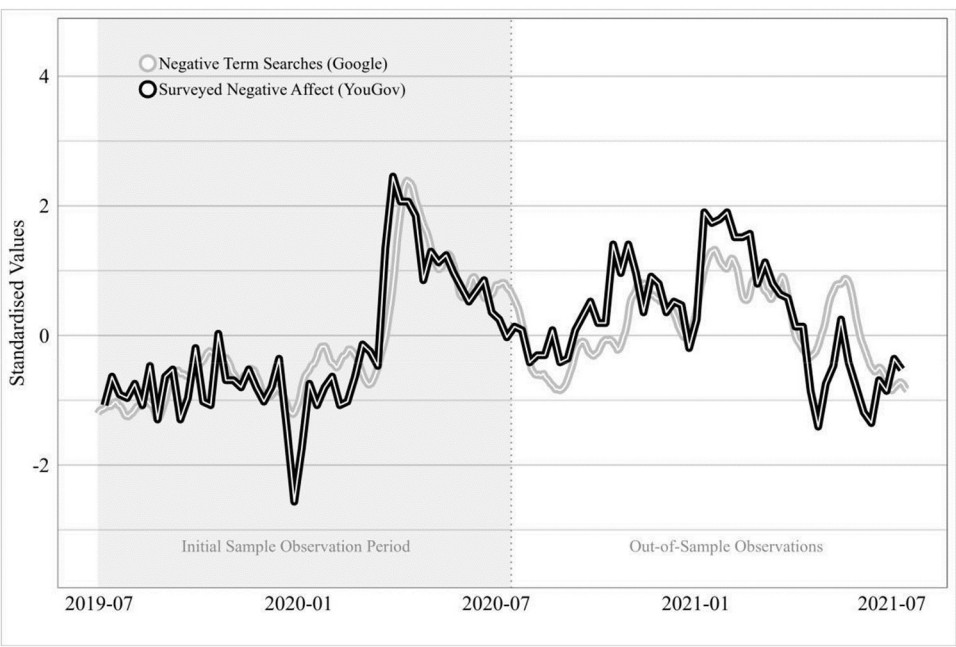

**Fig 3. Comparison of survey and Google trend series, June 2019 to June 2020.** The shaded (left) portion of the chart shows the correlation between negative affect mentions in the YouGov survey versus the negative affect search index based on Google data, at the time of the index construction (June 2020). The unshaded (right) portion of the chart shows how the YouGov weekly survey data and the Google search index have continued to covary in sync with one another during the subsequent year. The Negative Affect Index is based on YouGov weekly polling data, for a representative sample of circa 2,000 respondents across England, Scotland and Wales (216,441 total). It comprises the sum of all negative affect states reported by respondents. The Negative Affect Search Index is based on Google Trends data for the United Kingdom, and includes corresponding matches for stress ('psychological stress'), boredom, sadness, feeling scared ('fear') and apathy, weighted by their $R^2$ correlation with their individual matching terms. A two-week smoothing function has been applied to the weekly data for both measures. Indexes standardised (mean 0, standard deviation 1) for comparison purposes.

Having constructed and validated a negative affect search index for the UK, we then compare UK trends with those in other parts of the world. These comparisons are shown in Fig 4, which displays trends in the negative affect search index in the UK together with a broader range of English-speaking countries: Ireland, Canada, the United States, Australia and New Zealand.

Fig 4 shows that the trend observed in the British weekly survey data–of a sharp decline in affect before the lockdown as the COVID-19 pandemic accelerated, followed by a steady recovery after lockdown measures were put in place–is replicated across a wide variety of English-speaking countries globally. All cases experienced a spike in negative affect as the pandemic spread locally, and this appears synchronous with the country-specific timing of the outbreak. There is then an improvement in affect synchronous with the implementation of lockdowns, though this is more muted in 2nd and subsequent lockdowns, especially in Canada and Ireland. The only countries to avoid a renewed increased in negative affect during the period from late 2020 to mid-2021 were Australia and New Zealand, the two countries that implemented lockdowns that were successful in maintaining a zero Covid policy until their August 2021 wave (not shown). These countries also saw a much reduced spike in negative affect during their 2020 lockdowns in comparison to countries that experienced wide scale national epidemics. For example, in Australia negative affect rose to a peak of 2.5 times the pre-pandemic baseline

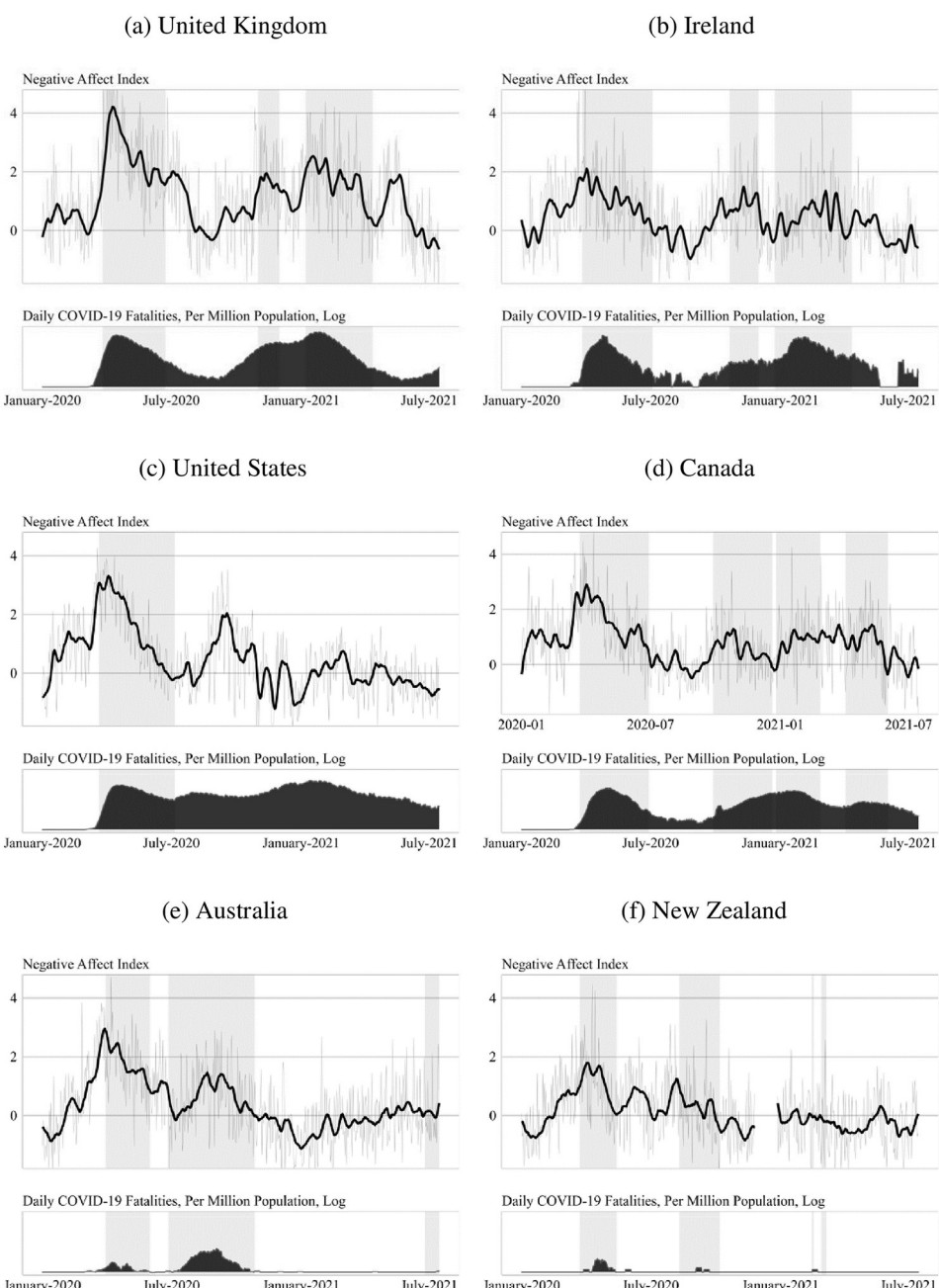

**Fig 4. Negative affect, lockdowns and pandemic intensity: Cross-country comparisons, January 2020 to July 2021.** (a) United Kingdom (b) Ireland (c) United States (d) Canada (e) Australia (f) New Zealand. Cross-country comparisons on the negative affect Google Trends index. All countries set relative to their pre-pandemic baseline period (15 January to 15 February). Shaded portions indicate periods of lockdown.

during the first lockdown in April 2020. In comparison, in the United States the negative affect index reached 3.3 times and in the United Kingdom 4.3 times the pre-pandemic baseline.

## Time-series cross-sectional models

In this section, we augment the clear descriptive trends outlined above with estimates of the association between lockdowns and mood from time series statistical models using the Google

Trends data. These allow us to control for some confounding factors. In particular, time series analysis allows us to address three issues. First, we must establish that the timing of the negative affect spike (and its subsequent decline) across countries is associated with the country-specific timing of coronavirus outbreaks. Second, we must demonstrate that the recovery of mood during lockdown is not wholly explained by the subsidence of the pandemic, to which lockdown was only one potential contributor alongside behavioural change and rising population immunity (though early evidence suggests immunity is mild at best [48]). Third, we need to show that the return to baseline during lockdown was more than a simple hedonic adaptation effect [10]–that is 'mean reversion' to set-point levels of good mood–as this too would imply that mood recovery was possible in the absence of lockdown measures. Such adaptation is a well-established phenomenon in the SWB literature [49].

We therefore estimate time-series models that control for the severity of the pandemic over time among countries for which comparative negative affect estimates can be calculated, as well as hedonic adaptation. Data on the severity of the COVID-19 pandemic is taken from the Johns Hopkins University Covid-19 Tracking Project [50]. As there is wide variation between countries and over time in the quality and effectiveness of COVID-19 testing, we use data on COVID-19 fatalities, which is less susceptible to measurement error. We follow a relatively new approach to estimating hedonic adaptation in the happiness economics literature, which is to include lag/s of the dependent variable, in our case the negative affective search index [51]. Most studies of adaptation instead use lags of the independent variable/s of interest, typically a shock like divorce, that the researchers are trying to estimate adaptation to [52]. This is inappropriate in our case because the independent variables of interest, namely pandemic severity and lockdown, are ongoing and varying over the period in question, rather than one-off events. We thus focus on whether the trends we observe can be explained simply by the general tendency of mood to adapt back to a baseline level over time [49]. The coefficient on the lagged negative affect search index can be interpreted within a difference-equation framework. If it is between 0 and 1, it implies that some portion of the past value of negative affect is carried over into the present period. The closer the coefficient is to 1, the longer this effect takes to decay, implying slower adaptation.

Models are estimated in the form:

$$NA_{c,t} = \alpha + \beta_1 NA_{c,t-1} + \beta_2 L_{c,t} + \beta_3 F_{c,t} + \beta_4 LD_{c,(t-l)} + \beta_5 E_{c,t} + \beta_6 ED_{c,(t-e)} + \mu_{c,t}$$

Where $NA_{c,t}$ refers to the negative affect index in time $t$ and country $c$, $NA_{c,t-1}$ to the one week lagged negative affect index in time $t$ and country $c$, $L_{c,t}$ to whether a country $c$ is in a lockdown period at time $t$, $F_{c,t}$ to the log number of daily fatalities per million in time $t$ and country $c$, $LD_{c,(t-1)}$ to the cumulative number of days $(t - l)$ since the onset of hard lockdown restrictions in country $c$, $E_{c,t}$ to whether a country $c$ is in a post-lockdown period at time $t$, $ED_{c,(t-e)}$ to the cumulative number of days $(t - e)$ since the easing of lockdown restrictions upon small businesses and retail, by country, and $\mu_{c,t}$ is the error term. All models are estimated using robust standard errors clustered by country, so as to account for serial autocorrelation, and also include both country fixed effects (not shown) and period fixed effects (by month of observation) to account for seasonal variation in subjective wellbeing [53].

We present the results of our analysis in Table 2. Models 1–3 are estimated using a longer sample period beginning in July 2019, well prior to the pandemic, whereas models 4–6 are estimated using a sample covering only the first eighteen months of the pandemic. Using two sampling windows in this way illuminates how the association between lockdown and mood differs statistically depending on whether it is assessed relative to a pandemic-free world (models 1–3) or to a world with pandemics but no lockdown (models 4–6). Columns 2 and 5

**Table 2. Negative affect under lockdown: Time-series models.**

| | Dependent variable: Negative Affect | | | | | |
|---|---|---|---|---|---|---|
| | Sample frame: | | | | | |
| | Since July 2019 (i.e. pandemic-free period) | | | Since February 2020 (i.e. post-pandemic but pre-lockdown) | | |
| | (1) | (2) | (3) | (4) | (5) | (6) |
| Daily COVID-19 fatalities per million, log | 0.136[†] | 0.11[†] | 0.117 | 0.194* | 0.167* | 0.166[†] |
| | (0.06) | (0.052) | (0.064) | (0.067) | (0.058) | (0.07) |
| Lockdown period (0/1) | 0.854* | 0.694* | 0.681* | 0.339 | 0.261 | 0.261 |
| | (0.226) | (0.17) | (0.191) | (0.244) | (0.198) | (0.205) |
| Days since lockdown start, log | -0.268* | -0.244* | -0.245* | -0.308* | -0.278* | -0.277* |
| | (0.078) | (0.066) | (0.062) | (0.086) | (0.07) | (0.069) |
| Post-lockdown period (0/1) | 0.181 | 0.109 | 0.100 | -0.365 | -0.350 | -0.349 |
| | (0.291) | (0.23) | (0.246) | (0.245) | (0.203) | (0.206) |
| Days since lockdown eased, log | -0.196 | -0.148 | -0.149 | -0.204[†] | -0.157[†] | -0.157[†] |
| | (0.106) | (0.082) | (0.086) | (0.092) | (0.07) | (0.072) |
| Negative affect index, lagged one week | - | 0.232** | 0.232** | - | 0.214* | 0.214* |
| | - | (0.049) | (0.05) | - | (0.053) | (0.053) |
| Log days under lockdown * log new fatalities (p. m.) | - | - | 0.000 | - | - | 0.000 |
| | - | - | (0.001) | - | - | (0.001) |
| Constant | 1.084* | 0.923* | 0.931* | 1.887** | 1.636** | 1.638** |
| | (0.364) | (0.279) | (0.265) | (0.33) | (0.234) | (0.232) |
| Observations | 4817 | 4768 | 4768 | 3341 | 3334 | 3334 |
| Adjusted $R^2$ | 0.218 | 0.26 | 0.26 | 0.255 | 0.29 | 0.289 |

Notes: All models use robust standard errors, clustered by country, together with fixed effects for country and month of survey to control for seasonal effects. Models are shown using two different sample frames.

all observations in the dataset (since July 2019), and b) all observations since the onset of the pandemic (February 2020).

[†]p<0.1;

*p<0.05;

**p<0.01;

***p<0.001.

introduce the lagged term that we use to estimate adaptation effects; columns 3 and 6 then add an interaction term for days under lockdown multiplied by new fatalities. Note that positive coefficients imply a worsening of negative affect, while negative coefficients imply an ameliorating association.

The model coefficients suggest the following inferences. First, the results support the view that country-specific pandemic severity was a major contributor to the elevated levels of negative affect observed during the period in question. The coefficient for log new fatalities per million is large and significant in all but model 3, such that a moderate increase in pandemic severity from 0 daily fatalities per million to 0.15 daily fatalities per million (as occurred in New Zealand or Australia) would raise estimated negative affect from 50th to 57th percentile of the distribution (0.18 standard deviations), whereas a much larger increase towards 10.5 daily deaths per million (as occurred in the United States) raises negative affect from the 50th all the way up to the 90th percentile (calculations derived from Model 4). These estimates are robust to the inclusion of the one-week lagged dependent variable for negative affect (columns 2–3 and 5–6), suggesting that levels of pandemic intensity can explain changes in affect even over relatively short periods.

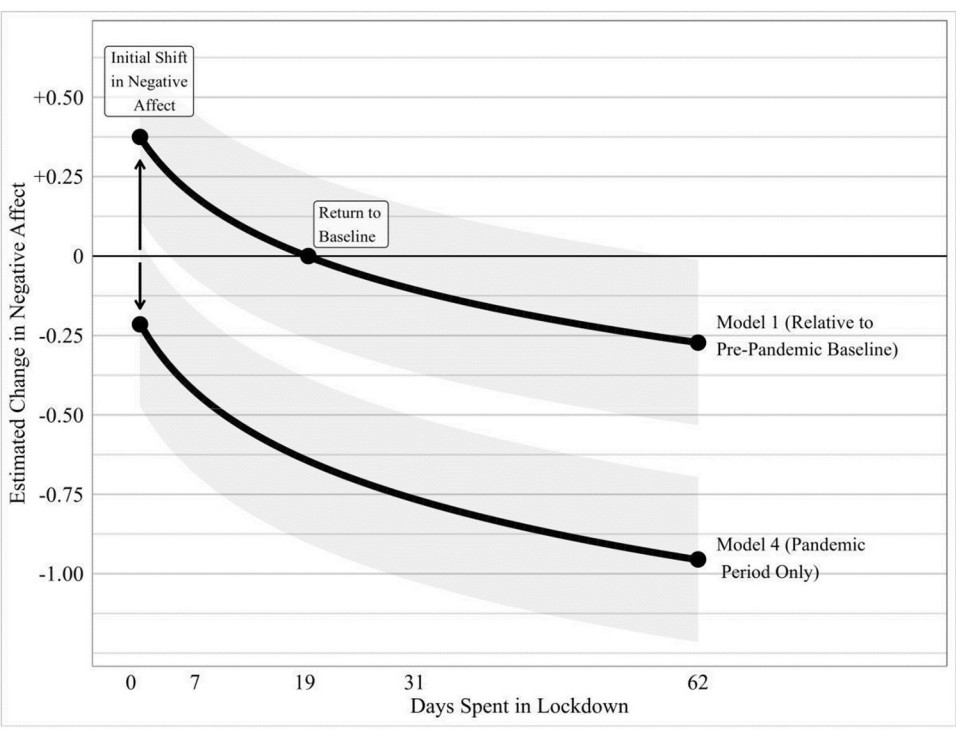

**Fig 5. Joint effect: Lockdown initiation and duration.** Controlling for natural mean reversion effects and the severity of the COVID-19 pandemic, negative affect is found to decline significantly during lockdown periods. Locally-estimated (loess) line of fit between points, with 95% confidence interval bound displayed. Regression coefficients used to estimate the component-plus-residual derived from Model 3.

Second, lockdown is statistically associated with a reduction in negative affect. Establishing this requires a comparison of the results from models 1–3 and models 4–6. In models 1–3, the dummy variable for being in lockdown has a large, positive, and significant correlation with negative affect. However, looking at the results more broadly, this appears to be because lockdowns are introduced amid pandemic outbreaks rather than because lockdowns themselves are associated with worsening mood. The coefficient for being in lockdown is not statistically significant if the sample frame is restricted to the pandemic period, as in models 4–6. This suggests that the result in models 1–3 is driven by comparison to a pandemic-free world. Lockdowns, however, are instituted in response to pandemic outbreak, so a pandemic-free world is arguably an unreasonable comparison case for the effect of lockdown. When the more reasonable comparison case of a pandemic outbreak is used, lockdowns do not seem to have deleterious impacts, broadly speaking. Indeed, the coefficient on days since lockdown began is large, negative, and statistically significant, suggesting that lockdowns reduce negative affect, as observed in our graphical analysis above. Fig 5 plots this ameliorating association between lockdowns and negative affect over time. While negative affect is above zero at the onset of lockdowns, this association attenuates to zero following a period of between 4 (Model 6) and 19 days (Model 1), where after negative affect falls below baseline. It is unclear why negative affect is above zero initially. It may simply be because deaths are spiking around this time. Or it could be an anticipation effect. Respondents might expect more deaths if the pandemic is severe enough to warrant a lockdown. They may also expect the experience of lockdown to be very unpleasant. As the reality of lockdown emerges, both in terms of its effect on death rates and its liveability, negative affect declines.

Taken together, these results suggest that, if anything, we should expect negative affect to improve following the implementation of lockdowns in response to pandemics. The decline in SWB observed in the studies we reviewed earlier would then be driven by the residual effects of the pandemic, which lockdowns have ameliorated rather than exacerbated. We observe no statistically significant relationship between transition to a post-lockdown period and negative affect. However, we note that days since lockdown eased is associated with improvements in negative affect, and is statistically significant at $\alpha = 0.1$ in models 4–6. This suggests that we should predict a negative association between lockdowns and affect in the absence of pandemic threat.

Finally, we find evidence of dynamics in negative affect that suggest adaptation, but this adaptation does not wholly explain the recovery of affect during lockdown. The coefficient on the one-week lagged term is significant but modest in size at 0.2. This implies that four- fifths of the effect of present circumstances decays within two weeks. Even with these variables in the model, days since lockdown remains significant and negative, while pandemic severity is significant and negative. This suggests that we have distinct forces in operation here, and the rebound in negative affect observed at the onset of lockdowns in our graphical analysis is not only the product of adaptation.

## Testing for group-specific effects: Multilevel models

The time series models in the previous section provide an overall picture of how affect varied in the population at large over the course of the pandemic and subsequent lockdown periods. This information is important to policymakers who are principally interested in the broad effects of the pandemic and lockdowns. However, these models obscure the heterogeneous effects of these events on various sub-groups. In our earlier study [46], we used the YouGov weekly mood data to estimate multilevel models with random slopes and intercepts by week of observation for key demographic groups in Great Britain to glean insights into these heterogeneous effects. We examined sub-groups by age, gender, ethnicity, socioeconomic status, and other life circumstances. As the updated individual-level dataset provided by YouGov (up until December 2020) contains a more reduced set of demographic variables, here we focus specifically on a comparison of those aged over 65 and those aged between 18–25 as this provides an important robustness check to our main results.

A priori, we might expect people aged 18–24 to be relatively more affected by lockdowns relative to pandemic outbreak than those over 65. Younger cohorts would have faced the stress of distance learning and the deprivation of face-to-face socialization, notably in indoor social venues, that resulted from government-imposed policy restrictions while being less concerned by the personal health consequences of COVID-19, which has much higher mortality among the elderly. During the whole of 2020, only 124 British residents under the age of 30 were estimated to have died from COVID-19 infection, compared to over 60,000 among those aged 75 and above [54]. In contrast, elderly citizens would have been relatively more concerned about mortality risk and thus more relieved by the imposition of lockdowns. A comparison of these sub-groups thus provides an opportunity to explore whether lockdowns or the pandemic have a relatively stronger association with negative affect. If we see an association between pandemic outbreak and worsening mood for both groups but no such association for lockdown, for example, this would further support the hypothesis that lockdowns ameliorate the negative impacts of pandemic outbreak. Furthermore, if we see our main results mirrored in the specific experiences of those over 65, who are no longer in the workforce, this suggests that our main results are not driven by furlough and similar economic supports made available to individuals of working age.

Multilevel models are commonly used in longitudinal analyses where period-specific events or processes may alter the relationships between individual attributes and outcomes of interest [55, 56]. Our data is highly appropriate for this sort of analysis. With around 2,000 observations drawn from a nationally representative sample by age, gender, social grade and region per each of 50 observation weeks, we have sufficient variation within and between weeks to enable relatively complex model specification among combinations of fixed and random effects.

We estimate multilevel models according to the standard specification:

$$SWB_{ij} = \left( \beta_{0j} + X_{0j} \right) + \beta_1 A_{ij}$$

Where $SWB_{ij}$ represents the score of subject $i$ on the subjective well-being measure in period $j$, $X_{0j}$ denotes the random effects design matrix consisting of ones in the first column (corresponding to the estimation of random slope intercepts) and second-level variables in the other columns, subscript $B_{0j}$ to the set of random slope coefficients for each time period $j$, $A_{ij}$ to a matrix of first-level independent variables including a constant term, for which time-invariant coefficients are provided by the vector $\beta_1$.

Fig 6 shows both the sociotropic (or period effect) as well as the random slopes estimated for the elderly and youth cohorts, with statistical significance highlighted in white. The sociotropic effect is derived from the random intercept term for each of the data and mirrors our main results above. affective life satisfaction declines during the spread of each pandemic wave, but recovers following the implementation of lockdowns in 2020 and 2021.

Among elderly individuals (those aged 65 and above), the random slope coefficients by survey week suggest an especially strong negative association between pandemic severity and affect in each pandemic wave, over and above the sociotropic effect common to all demographic groups. Elderly respondents exhibited levels of affect that were 0.02–0.04 points above the societal baseline before the onset of the pandemic, consistent with a longstanding literature which finds that life satisfaction rises as individuals reach the latter stages of life [3, 4]. However, as the novel coronavirus spread globally during the initial months of 2020, the surplus affect of the elderly turned to a significant (-0.04, p < 0.001) deficit. This recovered immediately from the start of the first lockdown as the pandemic was steadily contained, and by August the affect of elderly UK respondents had returned to the societal baseline. It then declined again to reach a significant (-0.025, p < 0.01) deficit as the spread of the more infectious Alpha variant resulted in a second wave of infections in September and October. Notably, in both instances the affect deficit of elderly respondents peaked as the virus was spreading, then began to recover almost immediately following the implementation of lockdown measures. In contrast, the trend in affect for young (18–24) survey respondents is basically flat despite this demographic being, intuitively at least, the most perniciously affected by lockdowns. This lends further credence to our hypothesis that the depressed subjective wellbeing observed by studies comparing pre-pandemic and post-lockdown survey responses is driven by the deleterious effects of the pandemic rather than those of lockdowns. If lockdowns were exacerbating negative mood, we would see a more pronounced decline among young people. Our results suggest that we should predict declines in the general population's affect with pandemic outbreaks and improvements following lockdowns introduced in response to those outbreaks.

## Discussion and limitations

We have alluded throughout our analysis to the difficulty of making causal inferences, due to the fact that pandemic waves and lockdown measures occur synchronously and endogenously.

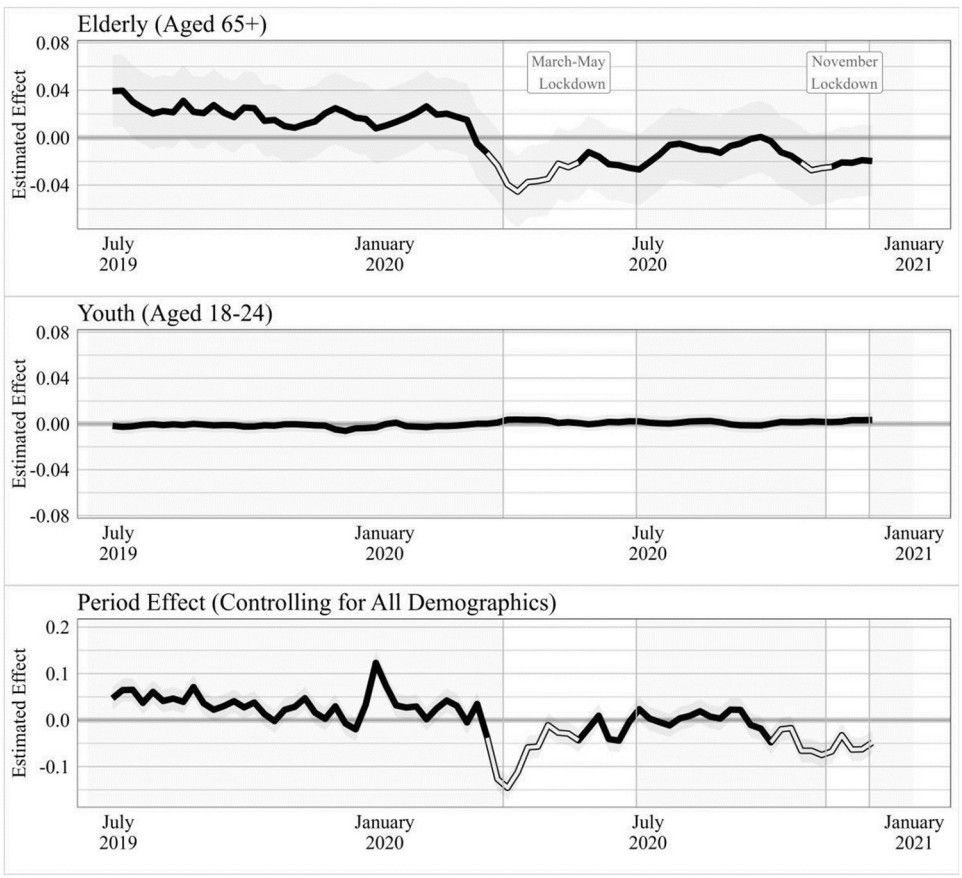

**Fig 6. Multilevel model random effects for key demographics, by survey week: Youth (18–24) versus elderly (65+) respondents.** Random effect slopes for socio-demographic variables, clustered by week of survey. Includes rolling average slope over the two prior and succeeding weeks. Note that individual demographic effects are over and above the period effect, and hence indicate that while youth wellbeing fluctuated in line with the national average, an especially sharp drop occurred among the elderly. 90% bootstrap estimated confidence intervals. Periods with statistically significant positive or negative effects highlighted.

Time series models allow only for the inference of (predictive) Granger causality rather than strict causal inference. However, the sample space of our current study does offer some note-worthy analytical leverage for sharpening our intuitions in the case of future pandemics. During the first wave of the global coronavirus pandemic, we observe that the increase in negative affect in New Zealand–a country that implemented lockdown without widespread community transmission–never rose more than 1.9 standard deviations above the pre-pandemic mean. Whereas in Great Britain, where a nationwide epidemic led to a significant number of COVID-19 fatalities, negative affect spiked to 4.3 standard deviations above baseline (p < 0.002). Similar patterns can be observed in Australia and the US, which had similar experiences to New Zealand the UK respectively, with affect worsening over the course of pandemic outbreaks but not over the course of lockdowns. Intuitively, this pattern suggests that the pandemic had a pronounced negative effect on affect, to which lockdowns were perceived to be an effective response, in particular when some of their negative side effects were mitigated through economic support measures. The nature of our method of analysing Google search data limits us to English-speaking nations. Extending our methodology to encompass a wider sample of countries would allow for the identification of additional cases where

lockdown policies and pandemic severity occurred in a divergent manner, as would within-country analysis taking advantage of subnational policy variation between U.S. states or constituent nations of the United Kingdom.

While we are unable to account for all mechanisms directly in our analysis, such as the burden of home-schooling, our results support the hypothesis that death rates are a major driver of trends in negative affect during conditions of pandemic and lockdown. Changes in death rates and affect mirror each other, broadly speaking, in all the countries we analysed. This suggests that lockdowns work on mood by reducing deaths, as substantiated by studies of excess mortality during COVID [1, 2]. An additional nuance to this view provided by our analysis is that the mood effects were most pronounced among the elderly, who were not affected by economic supports but were most at risk from COVID. During the first month of the pandemic, the proportion of young people (18–24) who reported feeling 'scared' during the past week to YouGov rose by 10 percentage points, whereas among the older respondents (65+) this figure was 19 percentage points. Lockdowns reduced deaths in this older demographic and, by association, reduced feelings of fear among this demographic and their loved ones.

There is at least one important confounding factor that we are unable to control for: the extent of socialisation during lockdown, both within and across households, potentially in ways that defied lockdown orders. People may have adapted their socialisation to suit lockdown conditions over time, such as by using video-call technology. This is consistent with the observed decline in feelings of loneliness after an initial spike at the start of lockdown. While we are unable to isolate the significance of these effects with our data, they do not alter our conclusions as they are inherent to lockdowns. However, individuals may also have returned to their normal patterns of socialisation with people from other households as lockdown went on. In that case, the rebound in affect would be a function of disobeying lockdown. Addressing this issue could be done using mobility data as a measure of the extent of voluntary isolation, as well as COVID-19 tracking surveys that ask questions regarding lockdown compliance.

A further issue is that our data is not longitudinal, in that the same individuals are not repeat-sampled across polling weeks, and so our results might be biased by sample variation. For example, people heavily affected by care burdens during lockdown may not have responded to the survey. This is unlikely as YouGov surveys are sampled so as to be nationally representative across survey weeks by age, gender, region and social grade, and the associated percentages are consequently broadly stable for the entire duration of the survey. Nonetheless, longitudinal studies would provide a complement to our analysis by further controlling for any potential selection bias across survey periods. Longitudinal studies that include niche subgroups of the population, such as those with pre-existing mental health conditions, would be especially complementary to our study in assisting policymakers when designing lockdown policies. Our results speak most clearly to the impact of lockdowns on the affect of the population as a whole, but people with certain characteristics may have acute experiences.

## Conclusion

Our results suggest that we should expect pandemic outbreaks to be associated with a worsening of affect across the population, especially among those most at risk. The more fatal and widespread the pandemic is, the more pronounced this worsening of affect will be. Furthermore, we should expect lockdowns introduced in response to such life threatening pandemics to be associated with an improvement in affect, at least in the medium term. While we do observe an increase in negative affect at the very beginning of lockdown, countries revert to baseline within 3 weeks at most, and thereafter see a net decrease. An intuitive explanation is that this is due to the mitigating effect lockdowns have upon the direct health impact of

infection and broader anxieties among vulnerable groups. Our results suggest that these trends are not entirely a function of adaptation to pandemic and lockdown conditions, nor can they be explained by furlough and other economic supports, as most pronounced trends are evident among those over 65 years of age who are out of the workforce.

## Supporting information

**S1 Appendix.**
(DOCX)

## Acknowledgments

The authors would like to thank YouGov for sharing the United Kingdom tracking poll survey data, and in particular Joel Rogers de Waal for his timely data updates and commitment to the project, and Stephan Shakespeare for the consistent support provided to the YouGov-Cambridge Centre for Public Opinion Research. We would also like to thank Annabelle Manley for excellent research assistance with the Google Trends data. We are grateful to Oxford's Wellbeing Research Centre for allowing us to present this work at their seminar series, and to attendees thereof for helpful comments. We are similarly grateful to two anonymous reviewers for their suggestions, which improved the manuscript substantially.

## Author Contributions

**Conceptualization:** Roberto Stefan Foa, Mark Fabian, Sam Gilbert.

**Data curation:** Roberto Stefan Foa, Sam Gilbert.

**Formal analysis:** Roberto Stefan Foa.

**Investigation:** Roberto Stefan Foa.

**Methodology:** Roberto Stefan Foa, Mark Fabian, Sam Gilbert.

**Project administration:** Mark Fabian.

**Validation:** Roberto Stefan Foa, Sam Gilbert.

**Visualization:** Roberto Stefan Foa.

**Writing – original draft:** Roberto Stefan Foa, Mark Fabian, Sam Gilbert.

**Writing – review & editing:** Roberto Stefan Foa, Mark Fabian.

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
