## [Decision Letter · Decision Letter 0]

25 May 2021

PONE-D-21-09713

Subjective Well-Being During COVID-19: Separating the Effect of Lockdowns from the Pandemic

PLOS ONE

Dear Dr. Fabian,

Thank you for submitting your manuscript to PLOS ONE. After careful consideration, we feel that it has merit but does not fully meet PLOS ONE’s publication criteria as it currently stands. Therefore, we invite you to submit a revised version of the manuscript that addresses the points raised during the review process.

We look forward to receiving your revised manuscript.

Kind regards,

Eugenio Proto

Academic Editor

PLOS ONE

Additional Editor Comments:

I have received two excellent reports, both reviewers found the paper very interesting and valuable, but both feel that the paper has to go through a substantial revision.

The main point they make is very similar: the main result cannot be interpreted as it is now, namely as the causal effect of lockdown on wellbeing, as a separated one from the general effect of the pandemic.

Furthermore, they both strongly recommend to tone-down the claim of causality. In the light of this the authors might also want to reconsider the title to some extent.

I therefore recommend a revision, where the authors should address each single point raised by the reviewers.

Journal Requirements:

3. Please ensure that you refer to Figure 2 in your text as, if accepted, production will need this reference to link the reader to the figure.

4. We note a previous version of your study was published by Bennet Institute for for Public Policy: https://www.bennettinstitute.cam.ac.uk/media/uploads/files/Happiness_under_Lockdown.pdf.

Please kindly clarify the following points:

a. Please clarify if the Bennet Institute for for Public Policy article was peer reviewed.

b. Please also clarify if the previously published article has been copyrighted. For your reference, PLOS ONE publishes all content under a CC BY 4.0 license (https://creativecommons.org/licenses/by/4.0/) which means that all material on our website is freely available online, and any third party is permitted to access, download, copy, distribute, and use these materials in any way, even commercially, with proper attribution. If the figures or text have already been published and copyrighted, authors must provide proper attribution, referencing the source clearly, and obtain permissions if the content is copyrighted.

Reviewers' comments:

Reviewer's Responses to Questions

**Comments to the Author**

1. Is the manuscript technically sound, and do the data support the conclusions?

Reviewer #1: Partly

Reviewer #2: Partly

2. Has the statistical analysis been performed appropriately and rigorously? 

Reviewer #1: Yes

Reviewer #2: Yes

3. Have the authors made all data underlying the findings in their manuscript fully available?

Reviewer #1: Yes

Reviewer #2: No

4. Is the manuscript presented in an intelligible fashion and written in standard English?

Reviewer #1: Yes

Reviewer #2: Yes

5. Review Comments to the Author

Reviewer #1: Report on “Subjective Well-Being during COVID-19: Separating the Effect of Lockdowns from the Pandemic”

This paper claims to separate the effect of the pandemic from lockdowns on subjective wellbeing using YouGov data for Great Britain in combination with Google Trends. According to the authors, the severity of the pandemic correlates negatively with subjective well-being (Table 2: positive adjusted “correlation” with the dependent variable NA (Negative Affect)) and the days under lockdown correlate positively with subjective wellbeing (Table 2: negative adjusted “correlation” with NA). The paper also shows that the relationship between lockdown and subjective wellbeing in the UK is heterogeneous (Figure 8): “the elderly, the affluent, and women living alone had especially negative experiences”, while “underemployed men saw a marked increase in their SWB during the lockdown” (p. 29).

The paper contains an empirical analysis on predictors of wellbeing using time series data, and it can speak of “Granger causality” at most, and not causality in a counterfactual sense (as the authors are well aware). I think the paper provides very valuable information, but the authors should reshape their presentation and discussion of findings (the key exhibits should be in the main text and the remaining ones in the appendix). As I argue below, I do not think it makes sense to talk about disentangling the effect of lockdowns from the effect of the pandemic (point 1). I also provide feedback on issues that need to be expanded and carefully revised (remaining points).

1. Causality vs. predictability

I have to say that I am totally baffled by statements such as “We are able to separate the effects of the pandemic from those of the lockdowns by utilising weekly data […]”. The lockdown does not occur in a vacuum, but as an endogenous response to an “exogenous shock”, the pandemic. Without clear assumptions stated either graphically (e.g. via DAG) or in equations (e.g. via structural models) is very difficult to assess how such an endeavour is even possible (https://journals.sagepub.com/doi/full/10.1177/2515245917745629). Given the way the authors present their findings, a natural question is: Should we expect a positive effect in the case of a “lockdown” without a pandemic?

Identification of causal effects: To claim that one can separately identify the effect of the “lockdown” (D) on subjective wellbeing (Y) from the effect of the “pandemic” (Z) seems a very strong, incredible claim. Any identification of causal effects must rely on assumptions.

One can identify the effect of the “pandemic” (Z) using a before-after strategy so long as seasonal effects accounted for (e.g. comparing Y before March 2020 vs. post March 2020 against before March 2019 vs. post March 2019); this is a “difference-in-differences” (DID) strategy with two time dimensions (year and month). The key assumption is that there are no other confounding factors changing in between periods (e.g., “parallel trends assumption”): in other words, the average change in Y would have been the same between pre-March 2019 and post-March 2019 that the one between pre-March 2020 and post-March 2020 in the absence of the pandemic. Unfortunately this paper does not have YouGov data before June 2019 (this is an important limitation of this paper).

But things are more cumbersome if one wants to identify the effect of the lockdown (D). For one thing, D is clearly a response to Z, so I wonder how one can identify the effect of D separately from the effect of Z. While one can think of Z as an exogenous shock, and try to get rid of seasonal effects via DID, D is an endogenous response to the pandemic. In this regard, it seems to me that the best one can hope for in this setting is to use Z as an instrumental variable for D and assume that the exclusion restriction is satisfied, i.e., that Z affects Y only via D. Effectively, then, one can at most think of identifying the effect of D thanks to Z. In practice, however, one will need to relax the exclusion restriction (e.g., https://www.jstor.org/stable/41349174)

In any case, if one is interested in causal effects, and disentangling causal effects, assumptions must be made, carefully stated, and discussed. This manuscript does not provide such causal inference approach, and it belongs to the realm of prediction studies.

Predictability (correlational) exercise: Do lockdowns predict subjective-wellbeing? One needs to be crystal clear about the fact that predictability is a complete different thing than causal inference (e.g. umbrellas don’t cause rain). I think the authors of this manuscript should be very careful in the way they present their findings and their interpretation of the empirical analysis in light of existing work. In general, the fact that a correlational analysis shows a strong positive relationship does not tell much, if anything at all, about the sign of the causal relationship between two variables. Moreover, things are quite complicated in the present setting, where there are important dynamics and different groups which generate a host of heterogeneities: the severity of the pandemic varies over time; its impact varies by time and (demographic) group; and the impact of the lockdown varies by time (or is a function of time) and (demographic) group, too.

The sentence “Alas, most empirical studies … of the pandemic” is confusing, since it seems the study will focus on causal effects, while indeed it will focus on correlations: “Were lockdowns associated with an improvement or worsening in subjective wellbeing?” Whether this is “essential input into contemporary and future policy debates” is debatable.

While the paper is in generally well written, there are a few paragraphs that generate a bit of confusion. On page 4 the authors write: “A shortcoming of these studies is an inability to distinguish empirically the effects of the pandemic from the effects of lockdown policies […] this confounds the effects of the two events. It is important that we identify the effects of lockdowns independently”. However, on page 2 the authors stress that “our results are primarily descriptive rather than causal”, so both the previous critique to existing work and arguing that “it is important that we identify the effects of lockdowns independently” are frankly problematic.

Many of the existing papers on the pandemic and mental wellbeing are well suited to identify the causal effect of the pandemic (under more or less explicit assumptions). However, the current paper claims to be descriptive and at the same time be concerned about papers which are able to identify well-defined causal effects (pandemic – including lockdown – effects). This requires a diligent revision and reformulation.

2. Previous research

Page 3 does not seem to provide a thorough review of existing work, omitting relevant references, of published (and unpublished) studies trying to understand the effects of the COVID-19 pandemic and the lockdown on mental wellbeing. While some papers focus explicitly on lockdowns, many others acknowledge that they cannot disentangle between the effects of the pandemic and the lockdown. The following studies seem relevant to the present paper:

Pandemic and mental health:

• Daly M, Sutin AR, Robinson E. Longitudinal changes in mental health and the COVID-19 pandemic: Evidence from the UK Household Longitudinal Study. Psychological Medicine.

• Daly M, Robinson E. Psychological distress and adaptation to the COVID-19 crisis in the United States. Journal of Psychiatric Research.

• Davillas A, Jones AM. The COVID-19 pandemic and its impact on inequality of opportunity in psychological distress in the UK. ISER Working Paper Series 2020-07.

• Ettman CK, Abdalla SM, Cohen GH, Sampson L, Vivier PM, Galea S. Prevalence of Depression Symptoms in US Adults Before and During the COVID-19 Pandemic. JAMA Netw Open.

Pandemic and mental health by ethnicity:

• Proto E, Quintana-Domeque C. COVID-19 and mental health deterioration by ethnicity and gender in the UK. PLoS ONE.

Pandemic and mental health by gender:

• Etheridge B, Spantig L. The gender gap in mental well-being during the Covid-19 outbreak: Evidence from the UK. Covid Economics 33.

• Oreffice S, Quintana-Domeque, C. Gender inequality in COVID-19 times: Evidence from Prolific participants in the UK. Journal of Demographic Economics.

Lockdowns and mental health:

• Adams-Prassl A, Boneva T, Golin M, Rauh, C. The Impact of the Coronavirus Lockdown on Mental Health: Evidence from the US. Human Capital and Economic Opportunity Working Group.

• Banks J, Xu X. The mental health effects of the first two months of lockdown and social distancing during the Covid-19 pandemic in the UK. Covid Economics 28.

• Niedzwiedz CL, Green MJ, Benzeval M, et al. Mental health and health behaviours before and during the initial phase of the COVID-19 lockdown: longitudinal analyses of the UK Household Longitudinal Study. Journal of Epidemiology & Community Health. 2020.

3. Interpretation of results

There are two key issues which are important in interpreting the results and which require additional work: compositional effects and anticipation (overshooting) effects.

The first relates to compositional effects. The data the authors use is cross-sectional, not longitudinal. This opens the possibility that the lockdown has a mechanical effect on the type of respondents. With the YouGov data it is possible to look at a bunch of demographic characteristics. The authors should plot the mean of different demographic characteristics of the respondents over time and look for patterns.

The second is about anticipation (or overshooting) effects. To what extent the correlation captured is not just reflecting an anticipation effect? In other words, people anticipate that a lockdown is coming, and the closer to the lockdown date the lower the reported mood (the higher Negative Affect) is. Indeed, very tough lockdowns were implemented in Italy (9 March 2020) and in Spain (13 March 2020). The earliest lockdown analysed by the authors is the one in the US (21 March 2020). Perhaps once people understand that the lockdown is not as tough as the ones in Spain and Italy, then mood improves. This is a different explanation for the findings documented in this paper, and requires further clarification and discussion.

4. Measuring the severity of the pandemic: cases vs. deaths

The authors should use alternative measures of the severity of the pandemic. Cases depend on whether widely testing was available (which varied across countries, in particular, early in the pandemic). The authors should present their analysis using deaths (instead of cases) too. Of course, there are also measurement issues with deaths, but it is another complementary way of looking at the severity of the pandemic.

5. Lack of data

It is unfortunate that the authors cannot replicate Figures 1 and 2 with data for January-June 2019 to account for seasonality effects (since the YouGov data are only available from June 2019). Nevertheless, and for completeness, it would be interesting to plot the data from June 2019 to June 2020.

6. Regression analysis

In addition to adding the analysis using deaths rather than cases, I think it is important to display the following set of results:

• Not controlling for lagged NA

• Controlling for country FE

• Controlling for month FE

• Interacting “days in lockdown” * severity of pandemic

• Interacting “days since easing of lockdown” * severity of pandemic

What is the interpretation of the coefficient on “days since easing the lockdown”? Is easing the lockdown a bad thing?

7. Other demographic characteristics

Do the authors have information on ethnicity and key worker status? I think it would be really interesting to see an analysis along these characteristics in Figure 8.

8. Data availability:

The authors write: "The raw data cannot be shared publicly as they are the commercial property of

YouGov. However, the time series of the data is published weekly here and is sufficient

to verify our analysis: https://yougov.co.uk/topics/science/trackers/britains-moodmeasured-

weekly. We will make all do files available to any researcher who wishes to

replicate our analysis." However, the link "" ext-link-type="uri" xlink:type="simple">https://yougov.co.uk/topics/science/trackers/britains-moodmeasured-weekly" does not seem to be working.

Minor comments

• Page 4, paragraph 1: it is not Good Health Questionnaire, but General Health Questionnaire.

• Page 4, paragraph 2: heterogeneous effects by demographic CHARACTERISTICS.

• Page 10 (OECD 2013). –space needed—All

• Figure 7: Does the lockdown starts when the line changes from “black” to “white” or is this indicated by the vertical line?

• There are 8 countries and 50 weeks of data: 400 observations. Column 5-6 display 376 observations, so 47 weeks of data are being used?

• Clustering standard errors with 8 countries is problematic. Wild Bootstrap should be used to ensure statistical inference with desirable properties: https://journals.sagepub.com/doi/full/10.1177/1536867X19830877

• Should the equations contain error terms?

Reviewer #2: This paper addresses an important and challenging topic within the context of the Covid-19 pandemic and shows how people’s mood changed in response to lockdowns during the pandemic. The paper’s relevance for policy is indisputable and the amount of data work that has been completed is noteworthy.

There are several major issues I’d like to draw attention to and it is very important that some of these are addressed given that the policy conclusions drawn from this research can influence people's lives.

1-It is not possible to accept the main argument that the paper distinguishes lockdown effects from the pandemic. The analyses are very rigorous but they cannot do much about the mere fact that lockdowns were introduced (and will always be introduced) in response to a crisis and therefore, there are no effects of lockdowns without the effects of pandemic. The results, therefore, are better interpreted as showing that the lockdowns attenuate some of the negative changes in mood that occurred during the pandemic, most plausibly by providing a sense of security and safety. This interpretation is supported by this basic intuition, as well as the main results reported in the paper which shows a rise in mood that follows from a reduction in mood. This interpretation is also supported by the corollary finding “that the seeming effectiveness of lockdowns in improving mood is conditional upon pandemic severity.” The authors also acknowledge this idea somewhere in the discussion by stating “ lockdowns improve SWB by ameliorating stress and fear associated with pandemic outbreaks. If there is no serious viral outbreak and thus little stress and fear to ameliorate, then lockdowns won’t improve SWB.” These arguments should be more central to the paper and dictate the title, abstract, and intro.

2-It is very important that the short timeframe for the estimates and the patterns in data that follow after this short-frame (how the data looks after the 1 month) are reminded to the reader in the abstract and throughout the manuscript. These are patterns observed early on during the lockdown and they mostly seem to disappear later on. Can the Figures 3 and 4 show later time periods in the graph to indicate this?

3-The authors should consider referring to ‘mood’ as opposed to ‘subjective well-being’ as this more accurately represents their data. Mood is a critical component of SWB and this link can be emphasized by citing literature and correlations with the life satisfaction measure but given that both of their outcomes measure mood (with some links to life satisfaction in one of the data), the paper seems to study mood and this should be reflected in the title, abstract, text etc. Relatedly, the efforts to link mood data to life satisfaction in the Yougov analysis doesn’t seem convincing. Using the lower quality life satisfaction data and so much imputation and complication doesn’t seem to be justified. Why not just use a simple index for positive and negative moods, and a composite mood index? The authors already use a negative affect index from Yougov data in their follow-up analysis and show trends that support their main finding in Figure 4, why not use this index (in addition to positive mood) in the main analysis too?

4-The negative affect trend in Figure 2 seems to indicate the complete opposite of the main findings throughout the manuscript. There are striking increases in boredom, loneliness, and apathy after the lockdown. This is very important and directly contradicts the main arguments in the paper. On the other hand, the reductions in stress and scare are very relevant and support the main mechanism that the lockdown increases a sense of security. Making these more central could enrich the theoretical contributions of the paper, which is currently not very strong. A follow-up question is: does this discrepancy in mood items emerge in the Google data too (i.e., results are different for stress, scare vs. boredom etc.) ? No matter if it does replicate in Google data or not, these diverging results need to be discussed very explicitly, they are important to acknowledge and they even help explain the results.

5-The causal language needs to be removed from the manuscript. Although the authors admit they are estimating associations, there is a heavy use of the words such as ‘effects’ or ‘impacts’. These words are better omitted.

6-The analyses do not completely control for the effects of the economic support from a causal inference point of view and the argument that the lockdowns always entail economic measures from a policy perspective is not convincing. It is possible that a stay-at-home order is not accompanied by economic measures and the degree of economic support can also change within a lockdown. I would recommend explicitly reporting this as a limitation. Authors can remind the findings for +65 adults as evidence that economic measures do not necessarily play a role, although it is possible that the economic support would contribute to the findings. Was there any way of controlling for this in the data, for example, by including or making references to the timeline of economic packages?

Minor comments

The discussion on Sweden in the intro is better placed in limitations.

1-I recommend that the following work on the topic and other recent studies that may have come out during the last months are integrated into the introduction and that the findings are discussed in light of this evidence:

Aknin, L., De Neve, J. E., Dunn, E., Fancourt, D., Goldberg, E., Helliwell, J., ... Amour, Y. B. (2021). A review and response to the early mental health and neurological consequences of the COVID-19 pandemic.

Giurge, L. M., Whillans, A. V., Yemiscigil, A. (2021). A multicountry perspective on gender differences in time use during COVID-19. Proceedings of the National Academy of Sciences, 118(12).

VanderWeele, T. J., Fulks, J., Plake, J. F., Lee, M. T. (2021). National well-being measures before and during the COVID-19 pandemic in online samples. Journal of general internal medicine, 36(1), 248-250.

2-"Underemployed men saw a marked increase in their SWB during lockdown" this could also be because of norm effects as more people become unemployed, unemployment could hurt less because there is less stigma, identity effects

3-Please temper the positive picture argument here “While our results paint a positive picture of the impact of lockdowns on SWB, “

4-Please provide more justification and explanation about why restricting sample space is necessary and what it entails “rising to -17% in models where the sample space is restricted to the period following lockdown onset (Models 3-4).” It is unclear what the sample in this sentence is.

5-The authors can be more accepting of the limitations of the cross-sectional nature of the data and present more information and discussion about the representativeness of both samples and whether and how the sample composition may have changed over time. This doesn't invalidate the importance of the findings but it is important to acknowledge and report in a study who has such strong population-level policy implications.

6-Are there any descriptive statistics that show how prevalent the specific population groups are in the data?

How is controlling for lagged values of the outcomes tackling hedonic adaptation? Please explain the mechanics/rationale for this new approach.

7-The negative trend in Figure 4 graph doesn’t match the results for Figure 2 ‘negative affect average’. Please acknowledge and/or reconcile this discrepancy.

8-The paper goes back and forth with the dataanalysis match, starting with YouGov data, then cross-country data, and then time-series analysis in cross-country data, and then finishing off with subgroup analysis in YouGov data again. It could simplify the paper if the authors start with the cross-country data (it only shows negative mood anyways) and time-series in this data, and then finish up with the Yougov analysis on negative and positive mood+subgroup analysis.

6. PLOS authors have the option to publish the peer review history of their article (what does this mean?). If published, this will include your full peer review and any attached files.

Reviewer #1: No

Reviewer #2: No

---

## [Author Response · Author response to Decision Letter 0]

15 Oct 2021

Please see attached response to reviewers

---

## [Decision Letter · Decision Letter 1]

22 Nov 2021

PONE-D-21-09713R1Subjective well-being during the 2020–21 global coronavirus pandemic: Evidence from high frequency time series dataPLOS ONE

Dear Dr. Fabian,

Thank you for submitting your manuscript to PLOS ONE. After careful consideration, we feel that it has merit but does not fully meet PLOS ONE’s publication criteria as it currently stands. Therefore, we invite you to submit a revised version of the manuscript that addresses the points raised during the review process.

If applicable, we recommend that you deposit your laboratory protocols in protocols.io to enhance the reproducibility of your results. Protocols.io assigns your protocol its own identifier (DOI) so that it can be cited independently in the future. For instructions see: https://journals.plos.org/plosone/s/submission-guidelines#loc-laboratory-protocols. Additionally, PLOS ONE offers an option for publishing peer-reviewed Lab Protocol articles, which describe protocols hosted on protocols.io. Read more information on sharing protocols at https://plos.org/protocols?utm_medium=editorial-emailutm_source=authorlettersutm_campaign=protocols.

We look forward to receiving your revised manuscript.

Kind regards,

Eugenio Proto

Academic Editor

PLOS ONE

Journal Requirements:

Additional Editor Comments:

The two reviewers have reacted very positively to the review, R1 only suggests that the author proofreads the manuscript, while for R2 some additional minor changes are necessary before the paper can be publishable.

Reviewers' comments:

Reviewer's Responses to Questions

**Comments to the Author**

1. If the authors have adequately addressed your comments raised in a previous round of review and you feel that this manuscript is now acceptable for publication, you may indicate that here to bypass the “Comments to the Author” section, enter your conflict of interest statement in the “Confidential to Editor” section, and submit your "Accept" recommendation.

Reviewer #1: All comments have been addressed

Reviewer #2: All comments have been addressed

2. Is the manuscript technically sound, and do the data support the conclusions?

Reviewer #1: Yes

Reviewer #2: Partly

3. Has the statistical analysis been performed appropriately and rigorously? 

Reviewer #1: Yes

Reviewer #2: Yes

4. Have the authors made all data underlying the findings in their manuscript fully available?

Reviewer #1: Yes

Reviewer #2: Yes

5. Is the manuscript presented in an intelligible fashion and written in standard English?

Reviewer #1: Yes

Reviewer #2: Yes

6. Review Comments to the Author

Reviewer #1: Thank you for preparing a careful revision. The revised paper is clearer and more effective.

I would suggest proofreading the article one more type for typos. For instance, on page 32, instead of . you should probably use : after "to control for":

"There is at least one important confounding factor that we are unable to control for. the extent

of socialisation during lockdown, both within and across households, potentially in ways that

defied lockdown orders."

Reviewer #2: I congratulate the authors for addressing the comments and rewriting their paper with valuable new analysis. The paper now more accurately captures people’s experiences during Covid-19 with new data, is stronger on potential theoretical and mechanistic explanations of the findings and literature review (in the introduction not so much in discussion) and uses a balanced language and adequately addresses the limitations.

A few important points remain to address - I believe these points can be addressed in writing by providing more justifications for methods and discussions of the findings:

1-The initial spike in negative affect at the onset of the lockdown --- how do authors explain this? I am not entirely sure if authors attribute this to the death rates or the lockdowns. It is possible to observe in graphical evidence that the Covid deaths also spike at exactly around this period (even in Australia and New Zealand, although to a lesser extent). It seems more plausible that this initial spike is a result of this jump in death rates as opposed to the introduction of lockdowns, but I can’t easily think of how this can be conclusively determined in the models. Given that the dummy for entering the lockdown period is not significant in models 4-6, it is plausible that the negative spike is not a lockdown effect. The robust relationship between death rates and affect could be evidence that the likely explanation for the initial spike is probably death rates, although it is imperfect evidence since it measures the general relationships with death rates at all levels and affects – not a spike. Given that there is already too many analyses in the paper, it would be difficult to ask for more analysis to explain this. At least, the authors can include these in the discussions – how do authors explain this spike? What do the analysis imply about the sources of this spike?

2- There is a very good discussion of the mechanisms in the literature review, but the authors don’t come back to this to discuss their findings. There is so much in the paper now about the potential mechanisms. Now with the mortality data, it is possible to see how death rates drop after the lockdowns in such a strong way and in the figures, the changes in death rates and the affect seem to mirror one another in all countries, which suggests that the declines in negative affect may be related to lockdowns stopping deaths. There are probably other papers showing this – that the lockdowns reduced Covid-related deaths? May be important to cite in the discussion. Then there is the evidence of elderly being the dominant group to carry the effects (how striking it is that the youth don’t show these changes), which suggests that lockdowns most probably reduced fear of death among those who fear the most OR the lockdowns decreased the rates of loved ones dying - which must be higher in older age populations. Again, it could be too much to ask for these explanations via analyses but these are some of the insights that can be at least discussed.

3-I think the strongest counterargument to the narrative of the paper is the hedonic adaptation effect- that Covid killed people those who stayed alive but at risk became sad and fearful people adjusted. I don’t believe this mechanism because the data is strong in showing how, in each instance, death rates dropped with lockdown and affect followed. But it is not possible to fully distinguish this. I do appreciate the authors using a new method to control for hedonic adaptation by controlling for the lagged DV. I understand that the coefficient shows the transmission of the well-being from one period to the next and indicates a timeline, yet, is this really adequate to control for hedonic adaptation? I can’t be sure. This is a novel method and I couldn’t find any example of this in the papers cited. In the papers cited, the controls are for lagged values of the independent variable (not DV). I can’t offer much insights into what the authors should do methodologically but I think they should be careful in this method/argument and triple check and explain how exactly that this method indeed controls for hedonic adaptation. The authors can also talk about what is the usual length of adaptation to bereavement or other death in the literature, if the adaptation occurs in shorter timeframes than what is observed with lockdowns, this can be used to support the findings. These are some suggestions to ease the doubts.

4- Some of the new analysis raise new questions, especially the split between pre-pandemic and post-pandemic in models 1-3 and 4-6 in the results. I present the questions that they raise below, it would be helpful to provide more answers to these questions in the manuscript.

• How is it that there was a period without the pandemic but with lockdowns? From July 19 till Feb 2020. Didn’t the lockdowns start only after the pandemic?

• What is the purpose of this distinction? What value does it add to the paper? “Using two sampling windows in this way illuminates how the association between lockdown and mood differs statistically depending on whether it is assessed relative to a pandemic-free world (models 1-3) or to a world with pandemics but no lockdown (models 4-6).”

• It is unclear what authors mean in this sentence and why we need this finding/approach, please explain in the manuscript: “The coefficient for being in lockdown is not statistically significant if the sample frame is restricted to the pandemic period, as in models 4-6. This suggests that the result in models 1-3 is driven by com-parison to a pandemic-free world.” Please include more explanations and justifications for this analysis and findings.

• How is the evidence for youth supports the hypothesis as indicated by the following sentence – if anything, it reads like the result for youth is calling the hypothesis into question and requires more explanation of why the results don’t hold for youth: “In contrast, the trend in affect for young (18-24) survey respondents is basically flat despite this demographic being, intuitively at least, the most perniciously a affected by lockdowns. This lends further credence to our hypothesis that we should predict declines in the general population’s affect with pandemic outbreaks and improvements following lockdowns introduced in response to those outbreaks.”

This association between lockdowns and affect is robust to controls for hedonic adaptation [10] and progress in containing the virus outbreak.

Minor comments:

Intro

• Page 2: replace comma with period after “introduced,”

• Page 3: Unclear what the following sentence means with easing timetables: “where easing timetables were maintained, despite the onset of a new coronavirus wave (such as the United States in the summer of 2020, and the United Kingdom in summer 2021).”

Descriptive results:

• Figure 1 – is it possible to indicate the onset of the pandemic in the graph? Realize the graphs are already populated but it was not possible to detect the main argument in the text in the graph: “They then fell sharply during the virus breakout in March before reverting higher following the stay-at-home order.” It even looks like there was a decline with the onset of the lockdowns.

• The weighting of the affect states is very clear now and the value and benefits of this method is well-explained.

• Could the authors put some numbers to the following comment? How was this conclusion reached? What are the sizes and statistical significance of these changes? “Taking all negative affect items together, negative affect rose sharply with the outbreak of the pandemic, and then continued to rise, albeit much more slowly, after the imposition of lockdown.

• Is it possible to put any statics to this claim – what is the size of this difference: “These countries also saw a much reduced spike in negative affect during their 2020 lockdowns in comparison to countries that experienced wide scale national epidemics.”

• In Fig 4, the affect changes in the second and third lockdowns seems smaller than the firsts. In Canada and Ireland, hard to observe a decline in affect the second and third lockdowns on later. This may be worth mentioning and explained.

Results

• Table 2: within the table, can the authors indicate “Pandemic free period” for models 1-3 and “Pandemics but no lockdown” in models 4-6. Otherwise, hard to follow the results.

• Table 2: please put coefficient of interest in the top row, so it is the first coefficient readers can see and follow.

• Page 24 – 25: In the first paragraph after Table 2, when describing the results, please direct readers to the right column or model. This could be achieved by putting column names in parentheses (Column 1) for the corresponding model/result. Otherwise, hard to follow the results.

• Better to detach these two sentences to avoid confusion--- it reads as if the first sentence summarizes the second (which wouldn’t be correct), but it becomes clear later that the first sentence summarizes the full paragraph: Second, lockdown is statistically associated with a reduction in negative affect. In models 1-3, the dummy variable for being in lockdown has a large, positive, and significant correlation with negative affect.

Discussion

• The discussion and limitation section currently includes only limitations. Please provide an overall discussion of the findings and explain how the findings relate to or complement existing literature.

• Page 32: There is a period after “control for”: There is at least one important confounding factor that we are unable to control for. the extent of socialisation during lockdown, both within and across households, potentially in ways that defied lockdown orders.

7. PLOS authors have the option to publish the peer review history of their article (what does this mean?). If published, this will include your full peer review and any attached files.

Reviewer #1: No

Reviewer #2: No

---

## [Decision Letter · Decision Letter 2]

24 Jan 2022

Subjective well-being during the 2020–21 global coronavirus pandemic: Evidence from high frequency time series data

PONE-D-21-09713R2

Dear Dr. Fabian,

We’re pleased to inform you that your manuscript has been judged scientifically suitable for publication and will be formally accepted for publication once it meets all outstanding technical requirements.

Kind regards,

Eugenio Proto

Academic Editor

PLOS ONE

Additional Editor Comments (optional):

The referee is now satisfied, I am delighted to recommend the publication of this manuscript.

bests

Reviewers' comments:

Reviewer's Responses to Questions

**Comments to the Author**

1. If the authors have adequately addressed your comments raised in a previous round of review and you feel that this manuscript is now acceptable for publication, you may indicate that here to bypass the “Comments to the Author” section, enter your conflict of interest statement in the “Confidential to Editor” section, and submit your "Accept" recommendation.

Reviewer #2: All comments have been addressed

2. Is the manuscript technically sound, and do the data support the conclusions?

Reviewer #2: Yes

3. Has the statistical analysis been performed appropriately and rigorously? 

Reviewer #2: Yes

4. Have the authors made all data underlying the findings in their manuscript fully available?

Reviewer #2: No

5. Is the manuscript presented in an intelligible fashion and written in standard English?

Reviewer #2: Yes

6. Review Comments to the Author

Reviewer #2: The authors addressed all comments and feedback. I have no further revisions to request. Best of luck.

7. PLOS authors have the option to publish the peer review history of their article (what does this mean?). If published, this will include your full peer review and any attached files.

Reviewer #2: No

---

## [Editor Report · Acceptance letter]

26 Jan 2022

PONE-D-21-09713R2 

Subjective well-being during the 2020–21 global coronavirus pandemic: Evidence from high frequency time series data 

Dear Dr. Fabian:

I'm pleased to inform you that your manuscript has been deemed suitable for publication in PLOS ONE. Congratulations! Your manuscript is now with our production department. 

Kind regards, 

on behalf of

Professor Eugenio Proto 

Academic Editor

PLOS ONE